# Is There a Fourth Law for Non-Ergodic Systems That Do Work to Construct Their Expanding Phase Space?

**DOI:** 10.3390/e24101383

**Published:** 2022-09-28

**Authors:** Stuart Kauffman

**Affiliations:** Department of Biophysics and Biochemistry, University of Pennsylvania, Philadelphia, PA 19104, USA; stukauffman@gmail.com

**Keywords:** Newtonian Paradigm, Ergodic Hypothesis, Second Law of Thermodynamics, non-ergodic universe, Kantian Wholes, constraint closure, free energy cost per added degree of freedom, exponential or greater expansion of phase space, Fourth Law of Thermodynamics, Entropy can decrease

## Abstract

Substantial grounds exist to doubt the universal validity of the Newtonian Paradigm that requires a pre-stated, fixed phase space. Therefore, the Second Law of Thermodynamics, stated only for fixed phase spaces, is also in doubt. The validity of the Newtonian Paradigm may stop at the onset of evolving life. Living cells and organisms are Kantian Wholes that achieve constraint closure, so do thermodynamic work to construct themselves. Evolution constructs an ever-expanding phase space. Thus, we can ask the free energy cost per added degree of freedom. That cost is roughly linear or sublinear in the mass constructed. However, the resulting expansion of the phase space is exponential or even hyperbolic. Thus, the evolving biosphere does thermodynamic work to construct itself into an ever-smaller sub-domain of its ever-expanding phase space at ever less free energy cost per added degree of freedom. The universe is not correspondingly disordered. Entropy, remarkably, really does decrease. A testable implication of this, termed here the Fourth Law of Thermodynamics, is that at constant energy input, the biosphere will construct itself into an ever more localized subregion of its ever-expanding phase space. This is confirmed. The energy input from the sun has been roughly constant for the 4 billion years since life started to evolve. The localization of our current biosphere in its protein phase space is at least 10^–2540^. The localization of our biosphere with respect to all possible molecules of CHNOPS comprised of up to 350,000 atoms is also extremely high. The universe has not been correspondingly disordered. Entropy has decreased. The universality of the Second Law fails.

## 1. The Newtonian Paradigm and Second Law of Thermodynamics Are Foundational to Classical and Quantum Physics

The Second Law of Thermodynamics is the most well-established theory in classical physics. Disorder–entropy–tends to increase. Given the time reversibility of the fundamental laws of classical and quantum physics, the Second Law of Thermodynamics is widely held to be the Arrow of Time.

However, must it be true? The conceptual foundations of the Second Law are two claims: (i) The Newtonian Paradigm: the system is in a pre-stated and fixed phase space [1]; and (ii) The Ergodic Hypothesis: the system spends equal time in equal volumes of its phase space [2].

Both claims are central to classical and quantum physics. Here, is the Newtonian Paradigm [1]. First, state the relevant variables. For Newton, these are position and momentum. Next, state the laws of motion in differential form coupling the relevant variables. For example, for Newton there are his three Laws of Motion and Universal Gravitation. Third, define the boundary conditions of the system. These boundary conditions thereby define the pre-stated and fixed “phase space” of all possible combinations of the values of the relevant variables. Fourth, state the initial conditions. Finally, integrate the equations of motion to obtain the entailed trajectory of the system in its fixed phase space. For classical physics this is a trajectory. For quantum physics, on most interpretations, it is an entailed trajectory of a probability distribution.

Here, is the Ergodic Hypothesis [2]. The system spends equal times in equal volumes of its *fixed* phase space. The Ergodic Hypothesis abandons integrating Newton’s equations of motion for the *N* particles in the box. Given *N* particles, 6*N* numbers specify the simultaneous positions and momenta of all *N* particles in the *fixed* 6N dimensional phase space. Given the Ergodic Hypothesis, the system will spend equal times in equal volumes of the fixed phase space so will spend more time in macrostates with more tiny 6*N* dimensional microstates than in macrostates with very few tiny 6n dimensional microstates. Hence, disorder will tend to increase, and entropy, the logarithm of the number of microstates in a macrostate, will tend to increase. This is the Second Law and the proposed Arrow of time.

Under the Second Law, the *localization* of the system in its fixed phase space *tends to decrease.* It is of fundamental importance that the Second Law depend upon the Newtonian Paradigm of a fixed phase space.

Does the universality of the Newtonian Paradigm hold? There are increasing grounds for doubt. Smolin [1] points out the no mathematical object can entail the evolution of the universe because the entailing relation is timeless, but time and Now are real. A timeless mathematical object has a fixed phase space. If time and Now are real, there can be no fixed phase space. Therefore, the Newtonian Paradigm cannot be universal.

Devereaux and colleagues [3] have recently shown that no modeler inside the universe can have a complete model of the universe. If correct, this rules out a pre-stated and fixed phase space for the entire universe, hence rules out a Theory of Everything that entails the evolution of the universe: The Newtonian Paradigm, requiring a fixed phase space, is thus not universal.

Kauffman and Roli have shown that it is not possible to use any mathematics based on set theory to deduce the ongoing evolution of ever-new adaptations in the evolution of life [4]. The evolving phase space of the biosphere includes these ever-new adaptations. *Thus, the ever-changing phase space cannot even be deduced*. Therefore, there is no fixed phase space. The corollary is that there can be no Theory of Everything the allows deduction of all that can or will happen in a universe containing at least one evolving biosphere [5]. The universality of the Newtonian Paradigm again fails. The clear implication of Kauffman and Roli is that the phase space of evolving life is not fixed [4,5]. The vast increase in the abundance and diversity of life in the past 4 billion years hints, but does not yet prove, that the phase space of the evolving biosphere has increased.

## 2. The Universe Is Non-Ergodic

It has become clear for some time that the Universe is non-ergodic on time scales vastly longer than the lifetime of the universe [6,7]. Consider encoded proteins found in prokaryotic and eukaryotic cells. A typical protein has about 350 amino acids linearly arranged in peptide bonds. Then, consider a shorter protein with 200 amino acids. How many proteins of length 200 are possible? Assuming 20 different encoded amino acids, there are 20^200^ = 10^260^ possible proteins length 200 amino acids. Could all of these have been synthesized in the lifetime of the universe?

The shortest time scale is the Planck time scale of 10^−43^ s. The universe is 10^17^ s old. There are an estimated 10^80^ particles in the universe. If all these particles, ignoring space-like separation, were creating proteins length 200 on the Planck time scale, it would require the age of universe raised to the 37th power to make all these possible proteins just *once* [6,7]. Therefore, at the scale of complex organic molecules such as proteins with 200 amino acids, the universe is vastly nonergodic. In fact, the universe is not ergodic above about 500 Daltons [8]. It is essential to note that the universe will be non-ergodic—not reach equilibrium—on time scales much longer than the lifetime of the universe [6,7].

## 3. The Biosphere Has Vastly Expanded Its Phase Space

Another basis to doubt the universality of the Newtonian Paradigm comes from a recent analysis. General relativity and cosmology both assume the Newtonian Paradigm to be foundational. Cortês et al. [9,10,11] undertook the analysis of the complexity of a biosphere compared to that of the abiotic universe, in order to assess the implications of the existence of life in the universe on the “Price” for the Initial State of the universe.

The Price for the Initial State is the Past Hypothesis: Given; (i) the Newtonian Paradigm with its fixed and unchanging phase space; (ii) the Second Law in which disordered complexity increases; and (iii) the present complexity of the universe; then, *the Initial State* of the universe must have been of correspondingly very low entropy. The present complexity of the *abiotic* universe is estimated to be an enormous $e^{10^{124}}$. The entropy of the Initial State was then the reciprocal: 1/$e^{10^{124}}$. Penrose points out how very difficult is this price for the initial state and the Past Hypothesis to be paid [12]. If the price *cannot* be paid, we have no past hypothesis, nor a cosmic arrow of time based on the Second Law.

The work of Cortês et al. [9,10,11] estimating the complexity of our single biosphere up to the first encoded protein synthesis makes this *price very much higher*. The price is not a “mere” reciprocal of $e^{10^{124}}$ but for a single biosphere *the price is the reciprocal of*
$10^{10^{237}}$, where *237 >> 124.* Because $10^{10^{237}}$ for our biosphere is vastly greater than $e^{10^{124}}$ for the entire abiotic universe, if we accept this vast number, we must conclude that the phase space of the biosphere has, in fact, expanded enormously. Critically, it is not necessary that the complexity of the biosphere be as vast as above. If the complexity of the biosphere is clearly larger than that of the abiotic universe, then the biosphere has truly expanded its phase space, and cosmology faces major choices.

Here are our choices for a cosmology that includes biospheres, i.e., a “Biocosmology” [9,10,11]. First, we can simply choose to deny the results of Cortês et al. Second, we can accept the Cortês results and choose to maintain the universality of the Newtonian Paradigm and Second Law. We then preserve current cosmology. However, in doing so, we also preserve a required Price for the Initial State, and we agree to pay the now vastly higher price. Third, the Price of the Initial State is conditioned on truth of the universality of the fixed phase space of the Newtonian Paradigm and the Second Law. Yet, if the complexity of the biosphere is clearly greater than the complexity of the abiotic universe, then the phase space of the evolving biosphere is not fixed but has expanded. Because the Newtonian Paradigm and Second Law both demand a fixed and pre-stated phase space, as our third choice, we can choose to abandon either or both of the universality of the Newtonian Paradigm and the universality of the Second Law for the cosmological evolution of the universe.

This article considers a candidate Fourth Law of Thermodynamics for non-ergodic systems such as evolving biospheres that can do thermodynamic work to construct their own expanding phase spaces. In such an expanding phase space, perhaps astonishingly, as I show below, order can increase, in flat contradiction to the Second Law. *The system constructs itself into an ever-smaller region of its ever-expanding phase space*. More, the biosphere constructs itself into the ever -smaller region of its ever-expanding phase *without a corresponding disordering of the rest of the universe. Entropy really does decrease.* This will be the Fourth Law.

Given the three choices above, I here take the third choice: Abandon the universality of Newtonian Paradigm with its fixed and pre-stated phase space. There is a “conservative” way to abandon the universality of the Newtonian Paradigm. Claim that: With the onset of life, evolving biospheres create new possibilities that expand a phase space that is no longer fixed [4,5,9,10,11]. However, prior to the onset of life, the Newtonian Paradigm holds, from the Big Bang onward, with its fixed phase space. In this case, the problem of the Past remains. The complexity of the current abiotic universe is estimated to be $e^{10^{124}}$. The initial state must pay the price. It is localized to the reciprocal of $e^{10^{124}}$. Penrose’s dismay persists [12].

Remarkable independent grounds exist to support the “conservative” way to abandon the universality of the Newtonian Paradigm only at the onset of evolving life. Paul Davies, in 2004 published a paper entitled, “Emergent Properties and the Computational Properties of the Universe” [13]. Davies argues that any physical law must be implemented within the resources of the universe. Given a maximum rate of elementary operations, 2*E/**p*, and that time starts at the Big Bang, he concludes that “an upper bound for the total number of bits information that have been processed by all the matter in the universe is … 10^120^. Expressed informally, the existence of an emergent law in a system of sufficient complexity that its behavior could not be described or predicted by processing 10^120^ bits of information will not come into conflict with any causal closure at the microlevel.” Taking account of Dark Energy, Davies raises the limits to about 10^122^. Davies concludes that proteins longer than 60–90 amino acids, and nucleic acids longer than 200 nucleotides are open to emergent behavior not determined by any causal closure at the microlevel. Finally, Davis notes that many proteins are far longer than 90 amino acids and many genes are far longer than 200 nucleotides, so emergence is not ruled out [13].

Davies arguments are entirely consistent with the complexity of the abiotic universe found by Cortes et al.: $e^{10^{124}}$. Taken together, the arguments of Davies and of Cortes et al. support the claim that the *Newtonian Principle holds for the abiotic universe,* however the vastly greater complexity of the evolving biosphere, ${\mathrm{10}}^{10^{237}}$, now becomes strongly positive evidence for emergence beyond the Newtonian Paradigm with the onset of life.

More, the huge excess of $10^{10^{237}}$ versus $e^{10^{124}}$ suggests that emergence in the evolving biosphere has been extremely important. The remainder of the article suggests some of the reasons for this.

## 4. Thermodynamic Work Has Been Done to Expand the Evolving Biosphere’s Phase Space

Because the universe is non-ergodic above about 500 Daltons, most complex things will never exist. Yet, the human heart, 300 g and able to pump blood, exists in the universe. How can this have become true [7,8]? To discuss this broad topic, I must explore ten issues:I ask my physicist colleagues to consider the question above. How indeed have hearts come to exist in the non-ergodic universe? The basic answer is that life emerged and evolved. Hearts pump blood that sustains the life of organisms with hearts. Organisms with hearts have offspring that also having hearts. Those organisms whose hearts function better at sustaining the whole organism, have more offspring. Natural selection selects for improve hearts. Organisms with hearts evolve. Thus, hearts exist in the non-ergodic universe [7].Organisms are Kantian Wholes. A Kantian whole has the property that the Parts exist for and by means of the Whole. The human reader of this article is a Kantian Whole. You exist for and by means of your parts: your heart, liver, kidneys and other organs and cells. They exist for and by means of you, the Kantian Whole [7].The simplest example of a Kantian Whole is a Collectively Autocatalytic Set. Gonen Ashkenasy has a set of nine small peptides, 1, 2, … 9. Each peptide binds and ligates two fragments of the next peptide into a second copy of the next peptide. Peptide 1 catalyzes by ligation a second copy of peptide 2. Peptide 2 catalyzes by ligation a second copy of peptide 3, and so on around a ring such that peptide 9 catalyzes by ligation a second copy of peptide 1. The system is collectively autocatalytic. No peptide catalyzes its own formation [14] The Kantian Whole is the entire set of nine peptides that constitute its parts [7,14].The existence of Kantian Wholes in the non-ergodic universe permits a non-circular definition the “*function*” of a Part in the Kantian Whole. *The function of a Part is that subset of its causal properties that sustains the Whole.* The function of peptide 1 is to catalyze the formation of a second copy of peptide 2. If peptide 1 jiggles water in the Petri plate that is a side effect, not its function. The function of the heart is to pump blood, not jiggle fluid in the pericardial sac or make heart sounds. Functions are real in the universe. The function of the heart is why it exists in the universe [7].Living cells and organisms achieve the property of *constraint closure* [15]. This property lifts life, based on physics, above physics in entirely unexpected ways. To wit, work is force acting though a distance. Atkins points out that, “*Work is the constrained release of energy into a few degrees of freedom*” [16]. Consider a cannon, cannon ball at the base of the cannon, and powder between the base of the cannon and the cannon ball. The cannon is the constraint and is also a boundary condition. When the power explodes at the base of the cannon, the cannon, as the boundary condition, constrains the release of energy in the expanding gas to expand only along the bore of the cannon.The expanding gas does thermodynamic work on the cannon ball which is shot from the cannon. Without constraints on non-equilibrium processes there can be no work.A new question: At the Big Bang, there were no cannons. Where did the cannon come from? It took work to make the cannon! The Work-Constraint Cycle: No constraint, no work. No work, often, no constraint [17]. If thermodynamic work requires constraints, where do the constraints “come from”? “Constraint Closure” is a newly discovered and transformative concept that answers the new question, “Where do the constraints come from? [15,17]. To envision *Constraint Closure*, consider three non-equilibrium processes, **1**, **2**, and **3**. Furthermore, consider three constraints, **A**, **B**, and **C**. **A** constrains the release of energy in process **1** that does work to *construct*
**B**. **B** constrains the release of energy in process **2** that does work to *construct **C***. Furthermore, **C** constrains the release of energy in process **3** that does work to *construct*
**A** [15]! *Constraint-closed systems do thermodynamic work to construct the very constraints on the release of energy into the few degrees of freedom that then constitutes the work that construct the very same constraints* [15,17]. We construct our artifacts—windmills and locomotives. Remarkably, Constraint-Closed systems literally do thermodynamic work to construct themselves by constructing their own boundary conditions that then constrain the release of energy into a few degrees of freedom to do work. We see next that living cells—Kantian Wholes—construct themselves via constraint closure.*Kantian Whole collectively autocatalytic sets also achieve constraint closure* [7]. To realize this, consider Ashkenasy’s nine-peptide collectively autocatalytic set [14]. Each peptide binds the two fragments of the next peptide and acts as a ligase linking the two fragments together via a peptide bond into the larger peptide. Thermodynamic endergonic work is done in forming that peptide bond. The peptide acting as a ligase and binding the two fragments of the next peptide lowers the activation barrier to form the new peptide bond. Therefore, the peptide ligase acting as a catalyst is a constraining boundary condition that constrains the release of energy into the few degrees of freedom that constructs the next peptide. Each peptide is a constraint [7,15]. Because each of the nine peptides acts as ligases for the formation of the next peptide around the ring of nine peptides, the entire system is a Kantian Whole that is also collectively autocatalytic and ALSO achieves constraint closure [7,14]. Thus, a collectively autocatalytic set achieves constraint closure and does work to construct itself as it reproduces itself [7,14,15].It is of central importance to point out that *the Newtonian Paradigm leaves the “boundary conditions” unspecified*. The boundary conditions of all possible values of the relevant variables constitute the phase space of the system, thus *changing the boundary conditions changes the very phase space of the system*. Again critically, *changing the phase space changes what is possible!*Living cells as Kantian Wholes constructing themselves and carrying out thermodynamic work cycles undergo heritable variation and natural selection. New molecules such as new proteins come into existence and can constitute *new boundary conditions. These create new in the universe phase spaces with new in the universe possibilities.* The creation of new in the universe possibilities is now permitted: we are beyond the fixed, pre-stated and fixed phase space of the Newtonian Paradigm. These create adaptations that truly are novel in the universe of possibilities. *The evolving biosphere expands its phase space* [4,5,9,10,11,17]. Both before and after the invention of encoded protein synthesis, life was capable of heritable variations that created ever-new molecular, morphological, and behavioral adaptations. In turn these created ever-new niches for ever-new species. The species diversity of the biosphere has increased enormously despite small and large extinction events. Thus, the phase space of the biosphere has expanded. It is important to realize that selection acts at the level of the Kantian Whole, not its parts. Furthermore, therefore, selection is *downward causation*. What survives is that which is fit in the current environment. This conclusion comes in opposition to *S*. Weinberg [18], “*it is not true that all the explanatory arrows point downward to particle physics*.”The evolution of ever-new adaptations that expand the phase space of the biosphere *cannot be deduced*. Adaptations are “opportunities” or “affordances” seized by heritable variation and natural selection. A given protein in a cell now used to bind a ligand can also come to be used to carry a tension load, or to transmit an electron. An engine block can be used as a paper weight and its corners are sharp and can be used to crack open coconuts. *It is not possible to deduce from the use of an engine block as a paper weight that the same object can be used to crack open coconuts* [4,5]. Because the indefinite uses on any object, X, alone or with other things, cannot be deduced and cannot be listed, *no mathematics based on set theory can be used to deduce the evolution of the biosphere* [4,5].*The implication of all the above is that evolution is a propagating construction that cannot be deduced, hence evolution is not an entailed deduction*. No Law entails the evolution of the biosphere whose expanding (or contracting) phase space cannot be deduced [4,5]. As a consequence, *the evolving biosphere must lie entirely outside of the Newtonian Paradigm*. In short, life is based on physics but beyond physics. There can be no “Theory of Everything” for the evolution of a universe having at least one evolving biosphere [4,5,17].

## 5. A Statistical Mechanics of Non-Ergodic Systems with Expanding Phase Spaces

A preliminary issue here is, how can we define the phase space for a non-ergodic system with an expanding phase space that can expand its phase space on a time scale vastly longer that the age of the universe? The natural concept of *the phase space of such a non-ergodic system* is the count all the possibilities that *might have occurred at any time* “*t*”. Call this “*P_t_*”. In general, some *subset* of all the possibilities at time, *t*, *P_t_*, will have *actually* occurred. Call this actualized subset “*A_t_*”. At any time, *t*, the ratio, *R* = *P_t_*/*A_t_* measures current *non-ergodicity* of the system at time *t*. Conversely, the reciprocal, 1/*R*, measures the *current localization* of the actual system in its total possible phase space at time *t*.

The temporal variation of *P_t_*, *A_t_*, *R*, and 1/*R*, that is, the non-ergodicity and localization of the system as the total possible expands, (or contracts) is then *a candidate 4th law for non-ergodic systems that can expand their phase space* [9,10,11]. Progress toward a candidate Fourth Law requires some new theory for how *P_t_*, *A_t_*, *R* and 1/*R* vary with time. The TAP Process [9,10,11] is the first such mathematical theory.

## 6. TAP, the Theory of the Adjacent Possible in the Universe

For the chemical evolution of the universe, of life in our biosphere, of technology for the past 2,500,000 years, the process appears to be described by a new equation, the TAP equation [9,10,11,19,20,21]:(1)$$M_{t+1}=M_{t}+\sum_{i=1}^{M_{t}}\alpha^{i}\left(\begin{array}{l} M_{t}\\ i \end{array}\right),\;0\;\leq \;\alpha \leq \;1$$

In this equation, *M_t_* is the number of “things” in the system at time *t*. A “thing” could be a kind of molecule, a species in the biosphere, a tool in a technological system or even an idea. In this equation, choose an initial value of *α*, for example *α =* 0.9. Then, use *α* raised to the *i*th power.

The behavior of this process is striking and appears to or does predict three unrelated distributions [11,19,20,21,22].

**I. The number of “things**”. If the process starts with a rather small number of types of items, *M*_0_ = 10, and *α* << 1.0, and iterates, the number of types of items increases glacially for a long time then explodes upward in a characteristic “hockey-stick” pattern. In the continuous version the number of things reaches infinity at a finite time. The TAP process thus has a *pole*. This hocky-stick growth is faster than any exponential. It is approximately hyperbolic, (The discrete version does not reach infinity but explodes very rapidly [11,20,21]).

The TAP process predicts and seems likely to fit the chemical evolution of the increasing number of kinds of atoms and molecules in the universe over 13.8 billion years [22]. Confirmation of a predicted hyperbolic increase in chemical diversity is required and of obvious interest. The universe started with no atoms, then created all the stable atoms, then simple then complex molecules. The enormous chemical diversity of the Murchison Meteorite formed with the solar system five billion years ago [22], is suggestive of a hockey stick.

The TAP process does fit the glacial then explosive increase in the numerical diversity of life. Life started on earth some four billion years ago. Glacially, for more than two billion years, life remained single-cellular—bacteria, archea, eukaryotes [23]. A diversity of multicelled organisms arose in the Ediacaran some 750 million years ago [23], then the number of species and higher taxa exploded in diversity in the famous Cambrian Explosion 550 million years ago. Thereafter, the number of lower taxa, families, genera and species, has continued to increase [23]. We have had had no clear account of this hockey-stick pattern of an increasing total number of species. TAP seems to fit it very well.

TAP seems clearly to fit the multi-million-year glacial, then very recent hockey-stick explosion of the number of goods and services, of tools. 2.6 million years ago, Australopithecus had perhaps 10 crude stone tools [19,20]. Two and a half million years later, 40,000 years ago, Cro-Magnon in France had a few hundred tools [19,20]. The Bronze Age 3000 years ago had perhaps several thousand tools [19,20]. In the past two centuries, the numerical diversity of our tools has exploded to billions. We have had no account of the explosion [24]. TAP is our first account of this hockey stick pattern [19,20,24,25]. More, TAP predicts the glacial then hockey-stick explosion in the past two centuries of global gross domestic product over the past two thousand years. TAP almost surely predicts the glacial growth in global domestic product since the introduction of compound tools perhaps 300,000 years ago [19,20,21].

**II. Simple to complex.** We can also interpret *M_t_* to be the most complex thing produced at time t, for example having *M_t_* parts. Then, the TAP process predicts a glacial then hockey- stick explosive increase both in the number of items and their gradual then explosive differentiation into simple and more complex items [11,19,20,21,22,23].

The TAP process surely fits the *increasing*
*complexity* of atoms and molecules in the universe over 13.8 billion years from no atoms to atoms, to ever larger molecules [22].

TAP clearly describes the evolution of the *increasing complexity* of living species, glacially for two billion years then the vast increase in complexity in the Cambrian Explosion and since over four billion years [23].

TAP also clearly describes the glacial then explosive *cumulative evolution of complexity* of tools in our technology since *Australopithecus* 2.6 million years ago. *Australopithecus* had perhaps ten crude and similar stone tools. For hundreds of thousands of years, tool complexity barely increased [19,20]. More than two and a half million years later, Cro-Magnon tools ranged more widely—needle to atlatl. In the Bronze Age complexity ranged from needles to chariots. Our billions of tools today range in complexity from needles to the International Space Station [19,20,21,24,25]. TAP is our first mathematical account of this hockey-stick pattern.

**III. Descent distributions.** TAP predicts a third, unrelated distribution. Each item that arises in TAP may have 0 to some number of direct children and 0 to some number of further descendants, grandchildren and so on. Thus, for each item, the total number of its descants can be determined. From this, it is possible to derive the descent distribution for all the items. This is a power law, slope—1.0 to 1.3, depending upon parameters.

Remarkably, TAP predicts the power law descent distribution for over 3,000,000 patients filed in the us patent office from1780 to today. For each patent, its antecedents can be determined as “prior art” cited in the patent application, and the result is a clean power law slope—1.2 [21]. Here, the “things” are not molecules, but ideas. This distribution parallels the history of technological evolution.

TAP can hope to fit patent descent distributions because each single patent derives directly from one or more parents. these parents typically have only a single “progeny” as in TAP. It seems likely tap will be helpful in understanding phylogeny including widespread horizontal gene transfer. This remains to be tested. In chemical evolution two substrate two produce reactions require a modest generalization of TAP but should show similar behavior.

The fact that the TAP process seems to fit three different distributions, suggests that it is capturing something quite fundamental about the long-term evolution of complexity in the universe.


**IV. TAP is the First Candidate for a Fourth Law of Thermodynamics for non-ergodic systems.**


It is important to stress again that we seek to define a phase space for a non-ergodic system that can expand its phase space on a time scale longer than the age of the universe. The system will never be ergodic. We will do so by using TAP. *TAP itself is never ergodic, hence a useful model for non-ergodic processes.*

TAP allows us to compute the *Total Possible, Tp,* the subset of the total possible that constitutes the *Actualized Possible*, *Ap*, and the ratio of these, *R* = *T_p_*/*A_p_* at any time, *t*. The temporal evolution of *T_p_*, *A_p_*, and ***R*** constitute the candidate 4th law.

Using TAP and setting *α* = 1.0 yields the evolution of the *Total Possible* as a function of time. Setting *α* < 1.0 and fixed, yields the time evolution of the *Actualized Possible*. Thus, TAP allows the calculation of *T_p_*, *A_p_*, and *R* as a function of time.

## 7. The Fourth Law

*The Fourth Law 4 states that T_p_, A_p_, and **R all tend to increase** with time.* The Fourth Law, 4, is remarkable. It says that *T_p_*, *A_p_*, and ***R*** do, in fact, *tend to increase* with time. Hence, *localization, 1/**R**, also does tend to increase*.

We will show below that the temporal behavior of our evolving biosphere confirms this, the Fourth Law. Most importantly, *when localization increases the rest of the universe is not proportionally heated,* so ***entropy decreases.***

## 8. What Is the Free Energy Cost per New Added Degree of Freedom?

What is the free energy cost per new added degree of freedom? This is a new question. In the fixed phase space of the Second Law, no issue can come up with respect to the free energy cost per added degree of freedom. In the Fourth Law, the phase space does expand, thus it becomes relevant to ask the free energy cost per added degree of freedom. The free energy *cost* per added degree of freedom should be *roughly linear* in the mass of the new thing constructed.

Consider a biosphere where the longest peptide length is *N*. Let a slightly longer peptide length *N* +1 be created. One new peptide bond has been created so the expansion of the phase space has a free energy cost of creating a single new peptide bond. *The free energy cost is, at most, roughly linear in N.* Further, it is well established that allometric scaling of ¾ power with respect to mass exists across 27 orders of magnitude for many phyla. The number of heartbeats per lifetime is independent of the mass of the organisms. *Here the free energy cost per added degree of freedom is independent of mass* [26].

Consider a biosphere whose longest peptide in at time *t* is *N*. The phase space is 20*^N^*. Let a new peptide length *N* + 1 arise. The new phase space is 20^(*N*+1)^. *The phase space has expanded exponentially. Importantly, this exponential expansion of the phase space required no more work and generated no more disorder/heat than that required to create the single longer peptide. It is critical that the disorder of the universe does not increase in proportion to the exponential expansion of the phase.*


*This result is a central conclusion: The phase space, R, increases exponentially. Therefore localization, 1/R, also increases exponentially, but the free energy cost per added degree of freedom is roughly constant. Therefore, **because the disorder of the universe does not increase in proportion to the increased localization, 1/R, the entropy of the total system truly decreases**. This flatly contradicts the Second Law.*


The claim is even stronger. In so far as the evolution of complexity in the biosphere and econosphere is characterized by the TAP process, its hockey-stick growth is *roughly hyperbolic with a pole, hence far faster than exponential*. In the hyperbolic hocky-stick phase of the process, adding a *single new element*, increasing Mt to Mt + 1, say 230 to 231, enables a new the set of possibilities: (Mt + 1 − Mt) >> 1.0. The expansion of the phase space is far greater than the disorder—heat added to the rest of the universe. Again, entropy decreases.

## 9. The Cost of Adding the Next Degree of Freedom Is Less as Degrees of Freedom Are Added

Under the Fourth Law, A new phenomenon emerges. The cost of adding the next degree of freedom becomes less as degrees of freedom are added. This is a major issue. For more than a century we have faced the question of how the evolving biosphere whose organisms do work cycles has managed to vastly increase in complexity in face of the Second Law. This is one of the major issues posed by Schrödinger in his famous book What Is Life. He asks if new laws of physics may be required [27]. It is true that living cells and organisms, by constraint closure, and work cycles do thermodynamic work to expand the phase space of the biosphere. If the Fourth Law, stating that the cost of adding each new degree of freedom becomes cheaper because the phase space is expanding exponentially or even hyperbolically is true, then evolving life does not have to overcome the Second Law. *Entropy in the universe is decreasing*. Can we test this with respect to our biosphere?

The Fourth Law makes a powerful prediction: At a constant energy input a system such as our biosphere can do work to expand it phase space will construct itself into an ever more localized subregion of its ever-expanding phase space, in other words, 1/*R* decreases. *Thus, order increases while entropy decreases.* Can this this true?

## 10. The Evolution of the Biosphere Confirms the New Fourth Law

Life emerged some 4 billion years ago. The annual energy input from the sun is roughly constant. Evolving organisms achieve constraint closure and do ongoing thermodynamic work to construct themselves in a propagating process by which an ever-constructed evolving biosphere of new possibilities keeps emerging [4,5,7,17]. The evolving biosphere does ongoing thermodynamic work to expand its own phase space.

The complexity of the biosphere is now enormous. However, that complexity is vastly smaller than all the possible biospheres than might have occurred*, **R**.* This is readily assessed by considering the just the known molecular complexity of the current biosphere considering only proteins of 2000 or more amino acids [28]. There are 20 raised to the 2000 power or 10^2600^ possible proteins of length 2000 amino acids. Thus, a realistic lower bound on the total possible phase space of our known biosphere today at the level of encoded proteins as legitimate physical degrees of freedom is 10^2600^. On this independent and data-based estimate, the complexity of the total phase space of our biosphere in terms of proteins as bound states is 10^2600^.

As Eigen suggested [29], perhaps $10^{60}$ actual proteins have been “tried” in 4 billion years. At constant solar energy input, the biosphere is vastly localized in the total molecular phase space, ***R***, it has constructed compared to the free energy cost to achieve this localization. Given Eigen’s estimate, the localization of our actual protein biosphere with respect to its total possible protein phase space is greater or equal to 10 ^60^/10^2600^ =/10 ^2540^.

The *free energy cost* to achieve the enormous localization above, 1/10^2540^, was the cost to construct 10^60^ proteins, *not to construct* 10^2600^ proteins. the biosphere is vastly localized in its phase space and the universe was not disordered to a corresponding extent. Entropy has decreased in the total universe because the phase space of the total universe has increased enormously at little free energy cost.

The claim of increasing localization treats proteins as the units of interest. Is this legitimate? Any given protein can be disassembled into its *N* atoms, then considered in a liter box of buffer and the standard 6*N* dimensional phase space. What is a protein? It is a specific macrostate that corresponds to the very small number of microstates consistent with the locations of its *N* atoms. The protein itself is highly ordered in its 6*N* dimensional phase space, largely a consequence of quantum mechanics and stable covalent bonds.

If we consider all the proteins in living organisms in the biosphere, and disassemble them into their total N atoms, this is a very large 6*N* dimensional phase space. The total macrostate of these N atoms assembled into all the proteins in the biosphere is localized in a tiny sub-volume of this very large 6*N* dimensional phase space.

Concentrating only on proteins ignores the diversity of possible complex organic molecules comprised of many atoms of carbon, hydrogen, nitrogen, oxygen, phosphorus, and sulfur, CHNOPS, in the actual biosphere. The largest encoded protein in mammals is Titin with 35,000 amino acids [30]. Each amino acid has on average ten atoms of CHNOPS. Therefore, Titin has roughly 350,000 atoms of CHNOPS. The phase space of all possible organic molecules of the current biosphere consists of all possible molecules comprised of 1 atom of CHNOPS, 2 atoms of CHNOPS, 10 atoms, 1000 atoms, 100,000 atoms, 350,000 atoms of CHNOPS. The total number of all possible molecules up to 350,000 atoms of CHNOPS is an unfathomably enormous number, *X*. The molecular diversity of our biosphere may be 10^15^ to 10^20^. Then, the localization of our biosphere in terms of organic molecules is 10^20^/X, which is an unfathomably small number. The disorder of the universe has not been correspondingly increased.

## 11. The Fourth Law Is Correct

If the 4th Law is valid and biospheres are abundant in the universe, the overall course of the total entropy of the evolving complex universe may need to be re-examined. Is the total entropy increasing or decreasing after life starts in a universe whose biospheres can do thermodynamic work to expand their phase spaces? Cortês et al. [9,10,11], estimate the complexity of our SINGLE biosphere to be $10^{10^{237}}$, vastly larger than the $e^{10^{124}}$ for the abiotic universe. Furthermore, 10^–2540^, and 10^20^/*X* are vastly localized. *If correct, the entropy of the universe really has decreased since the origin of life in the universe.*

The Fourth Law, 4, states that *T_p_*, *A_p_*, *R tend* to increase over time. However, the TAP process proceeds *inexorably* upward [9,10]. This is essentially unchanged if a first order loss term, –μ, per item at each time step is included [9]. An important limitation of the TAP process as a full version of a Fourth Law is that the variables created do not interact with one another. This is inadequate. In the evolving biosphere and global economy, species and goods create niches for one another. Species and goods go extinct, new species and goods emerge and flourish. Reasonable evidence suggests that both the evolving biosphere and econosphere are dynamically “critical”, and they generate power law distribution of extinction events in the Phanerozoic record, and power law distributed “Schumpeterian gales of creative destruction” in the evolving economy [31,32,33]. These dynamics appear to be endogenous. Because of these interactions among the items of TAP, the process *does not proceed inexorably* upward, but *merely tends* to proceed upward.

## 12. The Relation between the Fourth Law and the Second Law

The relation between the Fourth Law and the Second Law is straightforward. The Fourth Law reduces to the Second Law in the case of a pre-stated, fixed, and closed phase space that does not do work to expand its phase space.

## 13. We Have Taken the Second Law to be the Cosmological Arrow of Time—Is It?

If the reasoning and results noted above are correct, the Second Law is not Universal. It only applies to systems with fixed phase spaces. However, this is untrue of the evolving biosphere. More, the expanding phase space of the biosphere becomes part of the expanding phase space of the entire universe. We are sending spacecraft out of the solar system. The reasoning and results of this article imply that the Newtonian Paradigm is not universal. With the loss of that universality, the Second Law is not universal. The implication is that the Second Law cannot be a Cosmological Arrow of time that includes evolving biosphere

The further implication is that we need to rethink Cosmology itself. One approach to reconsidering Cosmology is to take non-locality as fundamental. The immediate implication is to flatly contract General Relativity with its locality [34]. Starting with non-locality as fundamental naturally yields a quantum gravity with a quantum arrow of time that is independent of the Second Law. This is not the interest of the present paper but may offer an alternative to the Second Law as the cosmological arrow of time [34]. This alternative may be testable [34].

## 14. Conclusions

Substantial grounds exist to doubt the universal validity of the Newtonian Paradigm that requires a pre-stated, fixed phase space. Therefore, the Second Law, stated only for fixed phase spaces, is also in doubt. The universe is not ergodic on vastly long-time scales. Living cells and organisms are Kantian Wholes that achieve constraint closure and do thermodynamic work to construct themselves. Evolution constructs an ever-expanding phase space. Thus, we can ask the free energy cost per added degree of freedom. That cost is roughly linear or sublinear in the mass constructed. However, the resulting expansion of the phase space is exponential or far faster. Thus, the evolving biosphere does thermodynamic work to construct itself into an ever-smaller sub-domain of its ever-expanding phase space at ever less free energy cost per added degree of freedom. Entropy really does decrease. A testable implication of this, the Fourth Law, is that at constant energy input, the biosphere can construct itself into an ever more localized subregion of its expanding phase space. This is confirmed. The localization of our biosphere in its protein phase space is at least 10^–2540^. The universe has not been correspondingly disordered. Entropy has decreased. The universality of the Second Law fails.

## Data Availability

Not applicable.

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
