# Peer review of "Is There a Fourth Law for Non-Ergodic Systems That Do Work to Construct Their Expanding Phase Space?"

_entropy, 2022, doi:10.3390/e24101383_

Round 1
Reviewer 1 Report
I have tried to understand the paper because it offers, at least by the title of the manuscript, an apparently avant-garde idea about a possible fourth law of thermodynamics that could contemplate the complex dynamics of evolutionary phenomena; and even affirming that it is put into doubt the Newtonian paradigm. However, in this attempt I have found a kind of copying and pasting of previous works by the author along with statements that do not have a detailed explanation of what is supported, not even simple examples to help understanding of the content of the manuscript for the benefit of the non-expert reader. In addition to these minimum requirements that are requested so that a research work does not seem merely speculative, I have found conceptual errors, misinterpretations, and remarks which I enumerate as follows:
1) Newtonian Paradigm is more known as mechanicistic paradigm not being the condition of a fixed space phase its main feature but only the assumption that things in the environment around humans are more like machines than like life.
2) Thermodynamics of living entities has been studied for instance, by Schrodinger and Brillouin, by means of cyclic processes. It would be expected some kind of discussion or comparison between the fourth law of the manuscript and the previous thermodynamic description for living organisms.
3) Contrarily to the author states, non-ergodicity is not in contradiction with Second Law but it only means a specific regime of the dynamics of a system. Take for instance, the Tsallis entropy S_q which can describe non-ergodic systems and at the same time allows to obtain the q-exponential distribution by maximization of the S_q entropy subjected to constraints. Here we see how the employment of the Second Law allows to deduce the q-exponential distributions which fit non-ergodic behavior.
4) I don't see the condition of a non-fixed phase space to be a problem or in contradiction with the Newtonian (mechanicistic) paradigm. In fact, an illustrative example of this is the logistic equation N'=rN(1-N/K) where N represents the population at instant t and r, K coefficients, clearly implies a variation of the degrees of freedom of the system, i.e. N = N(t), so the phase space describing the coordenates and momenta of the individuals is also varying. Thus, the deterministic character of this model, prescribed by N=N(t), can be compatible with a non-fixed phase space.
5) In lines 63 and 64: "The conceptual foundations of the Second Law are two claims: i) The 63 Newtonian Paradigm: the system is in a pre-stated and fixed phase space (1); and ii) The Ergodic Hypothesis: the system spends equal time in equal volumes of its phase space (2)." As far as I understand, the second law of thermodynamics is the result of empirical observations about the limitations of converting heat entirely to work, as stated by the equivalent postulates of Clasius and Kelvin, or also established by the efficiency of the Carnot engine. In other words, in its original form the Second Law does not refer at all the existence of a pre-stated fixed space phase but only the statement that it is impossible the total conversion of heat into work. Even more, we could think a chemical process in which starting from a single substance (product) it is generated two or more substances (reactants), with a the positive total variation of the entropy is positive and at the same time the phase space of the product and of the reactants result different in dimension (one initial component vs. two final components). With this example we see that the condition of a fixed phase space does not imply the validity or violation of the Second Law.
6) In lines 292 and 293: "The natural concept of the phase space of a non-ergodic system is the count all the possibilities that might have occurred at any time “t”."Again, what defines if a system is ergodic or not is not the phase space but only the correlations (expressed by the corresponding measure) between subsets evolved in time. More precisely, let us recall the definition of ergodicity of dynamical systems: "For every two sets A,B of positive measure of the phase space \Gamma, there exists n>0 such that \mu((T^{-n}(A))\cap B)>0, being \mu(E) the measure of E and T:\Gamma->\Gamma an invertible transformation."
7) In line 381 it is said: "Therefore, entropy truly decreases. This flatly contradicts the Second 381 Law." But the entropy of the environment could be compensating this effect in such a way the net result to be an increasing of the total entropy. This is not discussed in the manuscript.
8) In the manuscript it is not explained the connection of non-ergodicity and the TAP equation, assumed as the first mathematical description of the Fourth Law.
9) In line 345 it is presented the Fourth Law by means of two kind of axioms or conditions 1 and 2. I feel that this presentation is given in a criptical form for the non-expert reader because it refers to the sets A_p, T_p and the ratio R, being A_p and T_p not previoulsy defined. The only sets defined are A_t and T_t. Please it would be of great benefit for the reader if the conditions 1 and 2 are expressed in a self-contained form.
10) By the way, what is the need to formulate a Fourth Law? Assuming that this represents a generalization or correction of the supposed invalid Second Law in non-ergodic regimes (which we have seen by previous remark 3) that this incompatibility does not always occur), it would not be more properly to call it a generalized Second Law? It is tacit that a complemetary or new law must be include the older law as a particular case, in this case the Second Law.
11) The paper continues repeatedly with sentences and affirmations without the pertinent clarifications/discussions that make difficult to follow and appreciate its content.
Therefore, I recommend a major revision to amend the misinterpretations and clarify the novel concepts of the manuscript by taking into account the above remarks, in order to have a chance for publication.
Author Response
SEPT 16, 2021
I wish to express gratitude to the first reviewer for his careful and helpful comments on my manuscript, “Is There a Fourth Law for Systems that do Work to Expand Their Phase Space?” submitted to Entropy. The issues raised has helped me clarify the discussion.
I shall copy the successive points raised, and respond to each in order.
1)Newtonian Paradigm is more known as mechanicistic paradigm not being the condition of a fixed space phase its main feature but only the assumption that things in the environment around humans are more like machines than like life.
I thank the reviewer. I am using the standard definition of the Newtonian Paradigm as specified by Lee Smolin, reference (1). This definition is central for my purposes.
2) Thermodynamics of living entities has been studied for instance, by Schrodinger and Brillouin, by means of cyclic processes. It would be expected some kind of discussion or comparison between the fourth law of the manuscript and the previous thermodynamic description for living organisms.
I fully agree and have added this material to the text.
3) Contrarily to the author states, non-ergodicity is not in contradiction with Second Law but it only means a specific regime of the dynamics of a system. Take for instance, the Tsallis entropy S_q which can describe non-ergodic systems and at the same time allows to obtain the q-exponential distribution by maximization of the S_q entropy subjected to constraints. Here we see how the employment of the Second Law allows to deduce the q-exponential distributions which fit non-ergodic behavior.
I agree with the reviewer. A system can be non-ergodic in a fixed phase space before it reaches equilibrium and Tsallis entropy is useful. Typically these consider systems whose behavior is non-ergodic on short time scales with respect to the life of the universe. I here consider systems that are non ergodic on vastly longer time scales that that of the universe. My central focus is on the expansion of the phase space itself. It would seem of interest if Tsallis entropy can be extended here.
4) I don't see the condition of a non-fixed phase space to be a problem or in contradiction with the Newtonian (mechanicistic) paradigm. In fact, an illustrative example of this is the logistic equation N'=rN(1-N/K) where N represents the population at instant t and r, K coefficients, clearly implies a variation of the degrees of freedom of the system, i.e. N = N(t), so the phase space describing the coordenates and momenta of the individuals is also varying. Thus, the deterministic character of this model, prescribed by N=N(t), can be compatible with a non-fixed phase space.
I hope I may respectfully disagree with the reviewer here. In the logistic equation above the entailed tractor plots the abundance of the variable, N, of the population in an open Cartesian Coordinate system of time on the X axis and the value of N on the Y axis. In this case, the time axis is open. I believe this is not a familiar phase space for time is a variable here. In a phase space, the time variation is given by the vector field at all points in the phase space.
.
5) In lines 63 and 64: "The conceptual foundations of the Second Law are two claims: i) The 63 Newtonian Paradigm: the system is in a pre-stated and fixed phase space (1); and ii) The Ergodic Hypothesis: the system spends equal time in equal volumes of its phase space (2)." As far as I understand, the second law of thermodynamics is the result of empirical observations about the limitations of converting heat entirely to work, as stated by the equivalent postulates of Clasius and Kelvin, or also established by the efficiency of the Carnot engine. In other words, in its original form the Second Law does not refer at all the existence of a pre-stated fixed space phase but only the statement that it is impossible the total conversion of heat into work. Even more, we could think a chemical process in which starting from a single substance (product) it is generated two or more substances (reactants), with a the positive total variation of the entropy is positive and at the same time the phase space of the product and of the reactants result different in dimension (one initial component vs. two final components). With this example we see that the condition of a fixed phase space does not imply the validity or violation of the Second Law.
I agree with the reviewer although these classical efforts attempted to acieve a closed thermodynamic system. I am basing my analysis on Boltzmann’s later formulation of the Second Law in terms of Statistical Mechanics in a closed thermodynamic system in a fixed phase space. With the familiar N particles this yields the familiar 6N dimensional phase space, and with the ergodic hypothesis it yields the familiar Second Law. I base my analysis on this standard example of statistical mechanics.
6) In lines 292 and 293: "The natural concept of the phase space of a non-ergodic system is the count all the possibilities that might have occurred at any time “t”."Again, what defines if a system is ergodic or not is not the phase space but only the correlations (expressed by the corresponding measure) between subsets evolved in time. More precisely, let us recall the definition of ergodicity of dynamical systems: "For every two sets A,B of positive measure of the phase space \Gamma, there exists n>0 such that \mu((T^{-n}(A))\cap B)>0, being \mu(E) the measure of E and T:\Gamma->\Gamma an invertible transformation."
I believe the reviewer is correct and that this applies to a fixed phase space in a non-equilbirium state. As the reviewer says this in this analysis the issue is not the phase space itself. His issue is correct but does not address my new concern: My central concern is with a system that does NOT have a fixed phase space and is therefore necessarily not ergodic. I wish is to understand the expansion of the phase spade itself and the free energy cost of that expansion. These issues cannot arise in a fixed phase space whether at equilibrium or not. Another way of seeing the issue is that a fixed phase space has a Poincare’ recurrence time. No such recurrence time applies to a phase space that is ever expanding. (Of course the phase space of the biosphere cannot epand if solar input or energy input in general ceases.)
7) In line 381 it is said: "Therefore, entropy truly decreases. This flatly contradicts the Second 381 Law." But the entropy of the environment could be compensating this effect in such a way the net result to be an increasing of the total entropy. This is not discussed in the manuscript.
I thank the reviewer. His point touches on the central topic of the entire article and I have now tried to make it far more obvious. In the current case, because the phase space is expanding exponentially or even hyperbolically at the addition of each single item at roughly a constant or linear free energy cost, the expansion of the phase space is not accompanied by corresponding heat or disorder increase in the now vastly larger phase space. And because the system is now localized very more precisely in that ever vaster phase space, the entropy of the entire system has, in fact decreased. This is the heart of the issue. Entropy can decrease for systems that do work to expand their phase space. It is all very strange, but true.
8) In the manuscript it is not explained the connection of non-ergodicity and the TAP equation, assumed as the first mathematical description of the Fourth Law.
I again thank the reviewer and make the issue clear. TAP is an example of an ever non-ergodic process, hence a candidate for a 4th law.
9) In line 345 it is presented the Fourth Law by means of two kind of axioms or conditions 1 and 2. I feel that this presentation is given in a criptical form for the non-expert reader because it refers to the sets A_p, T_p and the ratio R, being A_p and T_p not previoulsy defined. The only sets defined are A_t and T_t. Please it would be of great benefit for the reason.
Again I thank the reviewer. I have removed the first of the two, leaving only one version of the 4th law.
10) By the way, what is the need to formulate a Fourth Law? Assuming that this represents a generalization or correction of the supposed invalid Second Law in non-ergodic regimes (which we have seen by previous remark 3) that this incompatibility does not always occur), it would not be more properly to call it a generalized Second Law? It is tacit that a complemetary or new law must be include the older law as a particular case, in this case the Second Law.
Again I thank the reviewer and am truly grateful for the chance to explain. The 4th law does precisely reduce to the Second law in the case of a fixed phase space where no work is done by the system itself to alter that phase space.
11) The paper continues repeatedly with sentences and affirmations without the pertinent clarifications/discussions that make difficult to follow and appreciate its content.
Again, I thank the reviewer. I have added clarifications and discussions throughout the MS.
Kind Wishes, the Author

Reviewer 2 Report
In the article, Autor considers 'fourth law' of thermodynamics for non-ergodic systems such as evolving biospheres that can do thermodynamics work to construct their own expanding phase space. Author expresses doubts about the universal validity of the Newtonian Paradigm that requires a pre-stated phase space. Therefore the Second Law of thermodynamics, stated only for fixed phase space, is also in doubt.
The presented views are very controversial, nevertheless worth publication and broad discussions.
Author Response
RESPONSE TO REVIEWER 2
I am grateful for the support of reviewer 2, and hope my responses to review 1make the manuscript more useful.

Round 2
Reviewer 1 Report
In this new version, the author has improved the manuscript by addressing my comments. In this effort, these ones have been answered satisfactorily, so I consider the paper can be published as it is.